# Specific Mutations near the Amyloid Precursor Protein Cleavage Site Increase γ-Secretase Sensitivity and Modulate Amyloid-β Production

**DOI:** 10.3390/ijms24043970

**Published:** 2023-02-16

**Authors:** Ryota Suzuki, Haruka Takahashi, Chika Yoshida, Masafumi Hidaka, Tomohisa Ogawa, Eugene Futai

**Affiliations:** Laboratory of Enzymology, Graduate School of Agricultural Science, Tohoku University, Sendai 980-8572, Japan

**Keywords:** Alzheimer’s disease, amyloid beta (Aβ), amyloid precursor protein (APP), gamma-secretase, intramembrane proteolysis, *Saccharomyces cerevisiae*

## Abstract

Amyloid-β peptides (Aβs) are produced via cleavage of the transmembrane region of the amyloid precursor protein (APP) by γ-secretase and are responsible for Alzheimer’s disease. Familial Alzheimer’s disease (FAD) is associated with APP mutations that disrupt the cleavage reaction and increase the production of neurotoxic Aβs, i.e., Aβ42 and Aβ43. Study of the mutations that activate and restore the cleavage of FAD mutants is necessary to understand the mechanism of Aβ production. In this study, using a yeast reconstruction system, we revealed that one of the APP FAD mutations, T714I, severely reduced the cleavage, and identified secondary APP mutations that restored the cleavage of APP T714I. Some mutants were able to modulate Aβ production by changing the proportions of Aβ species when introduced into mammalian cells. Secondary mutations include proline and aspartate residues; proline mutations are thought to act through helical structural destabilization, while aspartate mutations are thought to promote interactions in the substrate binding pocket. Our results elucidate the APP cleavage mechanism and could facilitate drug discovery.

## 1. Introduction

The amyloid hypothesis was proposed over 30 years ago as an explanation for the pathogenesis of Alzheimer’s disease (AD) [1]. Amyloid-beta peptide (Aβ) is produced via sequential cleavage of amyloid precursor protein (APP) by β- and γ-secretases. The peptides accumulate in the brain, which causes AD [2,3]. β-Secretase cleaves the extracellular region of APP to generate a β-carboxyl terminal fragment (β-CTF). β-CTF is subsequently cleaved by γ-secretase in its transmembrane domain through two different protease activities [3]. First, endopeptidase-like ε-cleavage releases the APP intracellular domain (AICD), which produces Aβ48 or Aβ49. Subsequent carboxypeptidase-like γ-cleavage (i.e., trimming) cleaves the first fragment into tri- or tetrapeptides, thereby producing the major Aβ species composed of 38–43 amino acids (Figure 1) [4,5]. Long Aβs (Aβ42 and Aβ43) are more prone to aggregate, and are also more neurotoxic, than short Aβs (Aβ38 and Aβ40) [1,6]. An increased proportion of long Aβ species leads to accumulation of Aβ fibrils and formation of senile plaques in the brain [7,8], causing AD.

γ-Secretase is a protein complex comprising four membrane proteins: presenilin (PS1 or PS2), nicastrin (NCT), anterior pharynx-1 (Aph-1), and presenilin enhancer 2 (Pen2) [9]. PS1 has nine transmembrane domains (TMD1–9) and two catalytic aspartate residues in TMD6 and TMD7 [10,11]. Upon maturation of the complex, PS1 is cleaved in the loop between TMD6 and TMD7, thereby producing N-terminal fragments (NTFs) and C-terminal fragments (CTFs) [3]. γ-Secretase is responsible for the cleavage of many single-pass type I membrane proteins (e.g., APP, Notch receptors, CD44, and E-cadherin) [12]. Familial Alzheimer’s disease (FAD) mutations have been identified in APP and PS1, with most being found in PS1 (>150 have been reported in PS1 to date) [1]. APP was the first gene in which FAD mutations were reported, and 19 mutations have been reported to date [13,14]. In particular, the TMD of APP is a hotspot for FAD mutations, which are attributed to an increased proportion of relatively long Aβs [15,16,17,18]. In contrast, the A673T mutation identified in the Icelandic population was found to be protective against AD, and was associated with smaller proportions of long Aβs [19]. Therefore, APP mutations alter cleavage via γ-secretase; however, the detailed mechanism is unclear.

Cryo-electron microscopy (EM) of γ-secretase has revealed its cleavage mechanism. The TMDs of each subunit are arranged in a horseshoe shape, with the extracellular domain of NCT located on the transmembrane helix [20]. PS1 has a hydrophilic catalytic pore structure that can be accessed by water [20]. Cross-linking experiments between β-CTF and PS1 revealed that β-CTF initially interacts with PS1 NTFs, and then moves to PS1 CTFs for proteolysis [21]. The structure of γ-secretase complexed with APP and Notch has recently been elucidated, and provides a structural basis for substrate recognition [22,23]. PS1 forms a hybrid β-sheet with the substrate at the catalytic pore, which may pull the helix of the substrate to unwind the cleavage sites. The structure of complexed γ-secretase with chemicals also revealed the binding site of the γ-secretase inhibitor (GSI) and γ-secretase modulator (GSM) [24]. GSI binds in the catalytic pore of PS1, while GSM binds to TMD1 of PS1. In recent years, GSM has emerged as a potential therapeutic target for the treatment of AD, as it suppresses the production of long Aβ species [3]. These studies are expected to lead to the design of more specific GSIs and GSMs.

We previously established a γ-secretase reconstitution system in yeast, which does not possess APP or γ-secretase homologs [25]. Aβ production was assessed by transcription of the Gal4 reporter, which allowed identification of PS1 secondary mutations that restore the cleavage of PS1 FAD mutants [26,27]. PS1 secondary mutations modulated γ-secretase activity and suppressed the production of long Aβs [27]. Therefore, yeast cells are useful for screening mutations and chemicals that alter γ-secretase activity. In this study, we evaluated the cleavage of FAD mutations in the APP TMD and identified APP secondary mutations that restored the cleavage of T714I, which is an APP FAD mutation. Some secondary mutations were able to reduce long Aβ levels. The possible effects of each secondary mutation on the cleavage of APP are also discussed.

## 2. Results

### 2.1. Effects of APP FAD Mutations on γ-Secretase Activity

A yeast reconstruction system expressing γ-secretase and an APP-based recombinant substrate (C55-Gal4p) were used to analyze APP mutations. Recombinant plasmids were introduced into the yeast strain PJ69-4A, which carries *HIS3*, *ADE2*, and *lacZ* genes under Gal4p control. γ-Secretase was reconstituted with PS1, NCT, Pen2, and Aph-1-aL without detection tags. After cleavage of the substrate, Gal4p is released from membrane-bound C55-Gal4p and activates the transcription of *HIS3*, *ADE2*, and *lacZ*. Thus, protease activity can be monitored based on cell growth and β-galactosidase activity.

Cleavage activities in the APP FAD mutant were evaluated in the yeast system. β-galactosidase activity determined the degree of APP mutant cleavage quantitatively. The β-galactosidase activity of I716F was comparable to that of the wild-type; however, the activities of T714I, V715A, and V717I were significantly reduced compared to the wild-type (Figure 2a). These data imply that some FAD mutations reduce ε-cleavage of APP. Among these mutants, T714I displayed the greatest reduction in cleavage (Figure 2a). In media lacking histidine and adenine (SD-LWHUA), T714I showed reduced growth after three days of incubation, indicating the reduced cleavage. On the other hand, V715A, I716F, and V717I showed similar growth to the wild-type (Figure 2b). Similar results were obtained at two days of incubation. No effect of V715A and V717I can be seen, even though statistically significant effects on β-galactosidase activity can be seen (Figure 2a). We interpreted that the expression of the reporter genes, *HIS3* and *ADE2*, in these mutants exceeded the threshold for full growth.

### 2.2. Secondary Mutations That Restore Cleavage of the APP FAD Mutant

Secondary mutations that restored APP T714I mutant cleavage were isolated to study how mutations alter the cleavage activity of APP. For screening, γ-secretase was reconstituted with FLAG-Pen2 and Aph-1aL-HA with detection tags. Previously, we found that a carboxy-terminal HA tag on Aph-1aL reduced its expression and γ-secretase complex in yeast (Appendix A). Relatively low γ-secretase activity with Aph-1aL-HA was suitable for growth-based screening. With Aph-1aL-HA and APP T714I, no positive growth was observed (Table 1 and Appendix A). Residues located on the carboxy-terminal side of the ε-cleavage site were chosen for mutagenesis. Random mutations were then introduced at V721/M722 or L723 of APP T714I, using PCR-based mutagenesis and primers with random nucleotides. Nine secondary mutants were isolated by screening 3 × 10^3^ cells for V721/M722 and one secondary mutant was isolated by screening 6 × 10^2^ cells for L723 (Table 1 and Appendix A). These mutants did not grow in the absence of PS1, indicating that they were cleaved solely by γ-secretase (Table 1).

Analysis of β-galactosidase activity indicated that all secondary mutants significantly increased cleavage activity, compared to T714I (Figure 3a). The significant difference between WT and T714I changes from *p* < 0.0001 (Figure 2a) to *p* < 0.05, but this may be due to the data set with large error in V721P/M722D. Immunoblotting confirmed that the expression levels of mutant C55-Gal4p were similar to those of the wild-type cells (Figure 3b).

### 2.3. Secondary Mutations Enhanced the Cleavage of APP

The cleavage of APP with secondary mutations alone indicated that all mutants had significantly enhanced cleavage activity, compared to the wild-type cells (Figure 4a). Immunoblotting confirmed that the expression levels of mutant C55-Gal4p were similar to those of wild-type cells (Figure 4b). These results imply that the secondary mutations enhanced cleavage reactions by γ-secretase. The secondary mutations had common elements with proline mutants at V721/M722 (V721T/M722P, V721S/M722P, V721P/M722G, V721P/M722D, V721P/M722H, V721P/M722Y, and V721P/M722N) or aspartate mutants at M722/L723 (V721P/M722D, V721S/M722D, V721G/M722D, and L723D). These results imply that proline and aspartate mutations at these positions have an important effect on APP cleavage.

### 2.4. Proline and Aspartate Mutations Enhanced the Cleavage of APP

The proline and aspartate mutations were further assessed by measuring the β-galactosidase activity of single proline mutants (V721P and M722P) and single aspartate mutants (M722D and L723D). When combined with T714I (*T714I*/V721P and *T714I*/M722P), the activity of the single proline mutants was 160% and 370% greater than that of wild-type cells, respectively (Figure 5a). These results imply that the proline mutants restored the cleavage of T714I. Single aspartate mutants combined with T714I (*T714I*/M722D and *T714I*/L723D) similarly restored cleavage of T714I at relatively low levels, comparable to the activity of wild-type cells (Figure 5a). The activity of proline and aspartate mutants alone was also greater than that of the wild-type cells (Figure 5b). These results imply that a single mutation to proline or aspartate at _721_VML_723_ is sufficient to enhance the cleavage of T714I or wild-type APP.

The enhanced APP cleavage observed in aspartate mutants may result from the charge carried by the mutations, which alters the length of the transmembrane site. To test this hypothesis, M722 and L723 were mutated to charged residues other than aspartate. Glutamate and lysine were used, which carry a negative and a positive charge, respectively. *T714I*/M722E and *T714I*/L723E mutants exhibited significantly higher β-galactosidase activity compared to T714I (Figure 5c). The glutamate mutants M722E and L723E also exhibited significantly higher β-galactosidase activity compared to the wild-type cells, comparable to that of aspartate mutants (Figure 5d). These results imply that glutamate mutations can also activate APP cleavage.

Lysine mutations in combination with T714I showed diminished β-galactosidase activity (Figure 5c). The lysine mutant, M722K, displayed <50% of the activity of the wild-type cells, while L723K was as active as the wild-type cells (Figure 5d). These results indicate that lysine mutations did not activate cleavage of APP. Immunoblot analysis confirmed that the expression levels of mutant C55-Gal4p were similar to those of T714I (Figure 5e), as well as to those of wild-type cells (Figure 5f). Notably, the negatively charged residue, but not the positively charged residue, profoundly affected APP cleavage. The length of the TMD did not appear to be a critical factor for cleavage of these mutants.

### 2.5. Aspartate Mutations Facilitated Cleavage of APP by Enhancing Interactions with the Positively Charged PS1 K380 Residue

We then assessed the ability of the aspartate mutants to activate APP cleavage through ionic interactions with the γ-secretase catalytic subunit, presenilin (PS1). A recent cryo-EM study of the structure of γ-secretase-APP revealed that APP M722 and L723 formed a β-sheet structure with PS1 (Figure 6d) [22]. It has also been suggested that APP _721_VML_723_ can fit into a large–small–large (S1′–S2′–S3′) binding pocket in PS1 [5,24]. Studies indicate that APP M722 and L723 are located near the lysine residue in PS1 K380, which is a part of the S3′ binding pocket (Figure 6d). To analyze possible interaction between the negative charge of aspartate mutation in APP M722D or L723D and positive charge of PS1 K380, PS1 K380 was mutated to glutamate (K380E) and β-galactosidase activity was measured (Figure 6a,b).

Aspartate and glutamate mutations at APP M722/L723 activated cleavage of APP in PS1 wild-type cells (Figure 6a), but activity was reduced when combined with PS1 K380E (Figure 6b). The degree of activation observed with APP aspartate and glutamate mutations was low, none, or even negative with PS1 K380E, compared to PS1 WT (Table 2). Immunoblot assays confirmed that the expression levels of mutant C55-Gal4p and PS1 were similar to the WT levels (Figure 6c). These results imply that aspartate and glutamate mutations at APP M722/L723 facilitate cleavage by enhancing interaction with the PS1 K380 residue through the APP-PS1 β-sheet.

### 2.6. Secondary Mutations Restored T714I in Chinese Hamster Ovary Cells

We further analyzed APP secondary mutations in Chinese hamster ovary (CHO) cells. Wild-type and mutant APPC99 genes were introduced into CHO cells through lipofection and the levels of secreted Aβ were quantified using immunoblotting, as described previously [28]. CHO cells with APP T714I or T714I and secondary mutations contained similar amounts of C99 in the cell lysate (Figure 7a, middle panel). The T714I mutant secreted a lower total amount of Aβ compared to the WT (Figure 7a, lanes 2 and 3, Figure 7b), indicating that ε-cleavage was impaired. The T714I mutant with secondary mutations secreted higher levels of total Aβ (Figure 7a, lanes 3–7, Figure 7b). These results indicate that single proline and aspartate mutations at V721P, M722P, M722D, and L723D were able to restore ε-cleavage of the T714I mutant to levels similar to those of the WT.

Aβ40 and Aβ42 were quantified using two-site ELISA. The T714I mutant secreted less Aβ40 (Figure 7c, left panel) and similar amounts of Aβ42 compared to the WT cells (Figure 7c, right panel). These results were consistent with previous reports of APP FAD mutants [15,16,17,18,29,30]. The effects of secondary mutations varied between proline and aspartate. Proline mutations (V721P and M722P) increased Aβ40 levels but reduced the amounts of Aβ42 with respect to T714I (Figure 7c). Aspartate mutations (M722D and L723D) did not alter the amounts of Aβ40 or Aβ42 relative to T714I (Figure 7d). These results imply that proline mutations restored the trimming activity and reduced long Aβ42, while aspartate mutations did not. For more detailed discussion, it became necessary to analyze all secreted Aβ species, including short Aβs.

The proportions of Aβ species were assessed using immunoblot and urea/SDS-PAGE analyses at pH 8.45 (Figure 7e) or 8.65 (Figure 7f). We chose two different pH conditions, since there were nonspecific bands around Aβ40 at pH 8.45. The production of each Aβ species was confirmed by the addition of DAPT, which is a γ-secretase-specific inhibitor. WT cells secreted mainly Aβ38 and Aβ40, together with small amounts of Aβ42 and Aβ43 (Figure 7e,f, lane 1). The proportion of Aβ fragments were quantified (Figure 7g,h). As summarized in Table 3, short Aβs (Aβ38 and Aβ40) comprised 91.4% of the total Aβs; long Aβs (Aβ42 and Aβ43) accounted for the remaining 8.6%. T714I drastically reduced the amounts of Aβ40 compared to the WT (Figure 7e,f, lanes 1 and 2), while the proportion of long Aβs increased to 21.5% (Figure 7g and Table 3). As shown in Figure 1, two Aβ production lines exist: Aβ49 → 46 → 43 → 40 (Cleavage line 1) and Aβ48 → 45 → 42 → 38 (Cleavage line 2). T714I shifted the cleavage line substantially from 1 to 2 compared to the wild-type cells (Table 3). These results imply that T714I mutation reduced trimming activity and favored the Aβ48 → 45 → 42 → 38 production line.

Compared to T714I, *T714I*/V721P secreted more Aβ40 and Aβ43, but less Aβ38 and Aβ42, while *T714I*/M722P secreted more Aβ40 and Aβ43, but less Aβ42 (Figure 7e,f, lanes 2–4). The percentages of the long Aβs decreased to 5.7% in the *T714I*/V721P mutant and 14.2% in the *T714I*/M722P mutant (Figure 7g and Table 3). Therefore, proline mutations were highly associated with reductions in the proportions of long Aβs. Proportion of Aβ cleavage line 1 comprised 91.3% in the *T714I*/V721P mutant and 61.4% in the *T714I*/M722P mutant, closer to that of the wild-type cells (Table 3). In addition, band intensity at the Aβ39 [31] was reduced by the T714I mutation and restored in the *T714I*/V721P mutant to WT levels (Figure 7e, lanes 2 and 3). Furthermore, bands at the Aβ41 [32] were observed in *T714I*/V721P and *T714I*/M722P mutants (Figure 7e, lanes 3 and 4). These results suggest that the proline mutations not only enhanced ε-cleavage but also increased trimming activity, and shifted the Aβ production lines from Aβ48 → 45 → 42 → 38 to Aβ49 → 46 → 43 → 40.

In contrast, the introduction of aspartate mutations (*T714I*/M722D and *T714I*/L723D) increased the amounts of almost all Aβ species (Figure 7e,f, lanes 2, 5, and 6). The M722D and L723D mutations reduced the proportion of Aβ40 and increased that of Aβ43, resulting in increased long-Aβs percentages of 40.5% for *T714I*/M722D and 32.3% for *T714I*/L723D (Figure 7h and Table 3). The proportion of Aβ cleavage line 2 comprised 75.8 % in the *T714I*/M722D mutant and 73.7% in the *T714I*/L723D mutant, similar to T714I (Table 3). These results imply that while aspartate mutations enhance ε-cleavage, they also impair trimming activity.

### 2.7. Secondary Mutations Modulate the Cleavage of APP

APP containing only secondary mutations was analyzed in CHO cells. Similar levels of APP C99 were detected in WT and mutant cells, with no significant changes in total Aβ production observed (Figure 8a, lanes 2–6, and Figure 8b). These results imply that secondary mutations do not affect the ε-cleavage of WT APP proteins in CHO cells. Aβ quantification by ELISA revealed that V721P and M722P reduced the amounts of Aβ42 (Figure 8c), and M722D reduced the amounts of Aβ40, compared to those of WT cells (Figure 8d). In contrast, the L723D mutant secreted similar amounts of Aβ40 and Aβ42 to those of WT cells (Figure 8d). These results imply that the trimming activity of γ-secretase is affected by APP secondary mutations alone. Therefore, it was important to analyze all secreted Aβ species, as we had done with the T714I mutant.

Immunoblot analysis revealed that the proportions of other Aβ species also changed. V721P and M722P caused reduced secretion of Aβ42 compared to that of WT cells (Figure 8e,f, lanes 1–3). Both proline mutations (V721P and M722P) increased the proportion of Aβ40 and decreased that of Aβ38 (Figure 8g). In summary, V721P and M722P increased the proportion of short Aβs and decreased that of long Aβs, and increased the proportion of cleavage line 1 (Table 4). In addition, a band at the Aβ41 position was observed with M722P (Figure 8e, lane 3). The effects of the proline mutations on the WT controls were similar to those observed for the T714I mutants. These results imply that proline mutations increase trimming activity with shifting the cleavage position toward Aβ40. In contrast, M722D caused increased secretion of Aβ42 and reduced the levels of Aβ40, while L723D reduced the levels of Aβ38 and Aβ42 (Figure 8e,f lanes 1, 4, and 5). M722D had a smaller proportion of Aβ40 (Figure 8h). The respective Aβ proportions of L723D were similar to those of the WT controls (Figure 8h). In summary, M722D increased long Aβs and preferred cleavage line 2; in contrast, L723D was similar to WT and preferred cleavage line 1 (Table 4). M722D appears to inhibit trimming, as observed in FAD pathology. Taken together, the results show that the proline mutations V721P and M722P modulate the cleavage of APP to lower Aβ42 levels, while the M722D mutation mimics FAD mutations.

## 3. Discussion

FAD mutations affect γ-secretase-mediated APP cleavage by producing more toxic Aβ species. However, the functions of FAD mutations and interaction between APP and γ-secretase are poorly understood. To address this knowledge gap, we previously screened suppressor mutations for PS1 FAD mutants and discovered that suppressor mutations could modulate γ-secretase activity [27]. Furthermore, we revealed that specific mutations in PS1 and Aph1 activated γ-secretase activity [33,34]. In this study, using a reconstituted yeast system, we identified several mutations in APP that restored cleavage of the APP FAD mutant T714I. The protease activity of γ-secretase depends on endopeptidase-like cleavage (ε-cleavage), as well as carboxypeptidase-like trimming, which includes γ-cleavage [4,5]. T714I severely reduced ε-cleavage activity in yeast cells, while the secondary mutations restored APP cleavage. In addition, secondary mutations alone activated cleavage. In CHO cells, secondary mutations restored the cleavage rate of T714I to that of WT cells, and modulated trimming. These findings prompted investigation of the effects of each mutation on the cleavage sensitivity of APP.

Proline mutations were identified in APP at positions V721 and M722. According to NMR spectroscopy [35] and cryo-EM structures [22], these residues form α-helices and also adopt β-sheet structures with γ-secretase when APP is cleaved (Figure 9). Unwinding the substrate helix is important for intramembrane proteolysis [36]. Destabilization of the transmembrane helix by a proline mutation in the hinge region (APP G716) [37] or glycine mutations in the middle of trimming sections (APP I718 and T719) [38] strongly affects the cleavage of APP by γ-secretase. It was shown that proline mutations destabilize the α-helix and β-sheets, making them flexible [39,40]. The proline mutations identified in this study (V721P and M722P) are assumed to facilitate ε-cleavage and subsequent trimming activity by increasing the flexibility of the transmembrane helix.

Molecular dynamics simulations indicate that the transmembrane helix of T714I is shifted toward the C-terminus, compared to WT proteins [41]. The proline mutations isolated in this study may cancel out this shift. In addition, the proline mutations produced a specific band between Aβ40 and Aβ42 corresponding to Aβ41, which was not observed in WT cells. Disruption of the β sheet by proline mutations may affect the cleavage site. The two main pathways for Aβ cleavage are Aβ49 → 46 → 43 → 40 and Aβ48 → 45 → 42 → 38 (Figure 1). The introduction of proline mutations may result in a novel cleavage pathway, specifically Aβ50 → 47 → 44 → 41.

Aspartate mutations were identified at APP locations M722 and L723. As shown in Figure 6d, these residues were in close proximity to PS1 K380, which implies the presence of ionic interactions. Glutamate mutations may not have been identified during screening because their recovery levels may have been lower than those of the aspartate mutations. When we prepared cells with each replacement, T714I mutants in combination with the glutamate mutation (M722E or L723E) grew only partially, compared to the aspartate mutations (M722D and L723D) (Appendix A). Therefore, the growth difference of T714I mutants has a smaller effect on activity, making it difficult to isolate mutants during the screening of replicates. The PS1 K380 mutant was identified in a previous screening of suppressor mutants of the PS1 FAD mutant L166P [27], which indicates the importance of these residues. In addition, the structure of γ-secretase in complex with the transition-state analog γ-secretase inhibitor, L-685,458 [24], as well as the results of molecular dynamics simulations, show that PS1 K380 is located in the S2’ or S3’ pocket [42] and likely interacts with APP M722 or L723. Therefore, aspartate mutations may determine the preferred Aβ cleavage points.

In APP WT cells, ε-cleavage occurs at Aβ49 when the side chains of PS1 K380 and APP L723 face in the same direction, while those of APP M722 face the opposite direction (Figure 6d). Therefore, the ionic interaction of APP L723D and PS1 K380 leads to Aβ49 → 46 → 43 → 40 reactions, while the ionic interaction between APP M722D and PS1 K380 facilitates the shift of the APP transmembrane helix, which underlies the Aβ48 → 45 → 42 → 38 pathway and FAD pathology. The T714I mutant contained both M722D and L723D, which exacerbated FAD pathology. The mutation in the middle of the Aβ trimming section negatively affected T714I trimming, which cannot be ignored even if aspartate mutations determine the preferred ε-cleavage site.

Specific mutations that facilitate cleavage of APP were found in this study; V721P and M722P increased both ε-cleavage and trimming activity, decreasing long Aβ species. This supports the correlation between ε-cleavage and trimming activity, and also elucidates the cleavage mechanism of APP. The results in this study indicate that the yeast system is useful for analyzing mutant interactions between APP and PS1 mutations. We would like to extend the study with other PS1 mutations and clarify the substrate specificity of γ-secretase in the future. However, it still remains necessary to analyze the γ-secretase activity in mammalian cells using luciferase assay in PS1/PS2 DKO cells [27], introduced with PS1 and APP mutants. FRET-based fluorescence lifetime imaging to monitor APP-PS1 interactions is also suitable for analyzing the enzyme–substrate interaction in mammalian cells [43]. To date, the mutations identified in this study have not been found in human SNPs (gnomAD and jMorp; Tohoku Medical Megabank Organization) [44,45] or somatic mutations in cancer [46]. Recently, it was discovered that Verteporfin is a GSI that binds to the transmembrane site of APP [47]. Drug design has explored GSIs and GSMs that target APP, as well as γ-secretase [48,49]. Continuation of these studies will enable the design of more substrate-specific GSMs and other therapeutics.

## 4. Materials and Methods

### 4.1. γ-Secretase Reconstitution and Reporter Assays

γ-Secretase was reconstituted in yeast using human PS1, NCT, FLAG-Pen2 or Pen2, Aph-1aL-HA or Aph-1aL, APP-based substrate C55 (amino acids 672–726 from human APP770), and Gal4, which were cloned into their respective vectors as described previously [26]. Briefly, PS1 and NCT were cloned into the *Kpn*I and *Xba*I sites of pBEVY-T [50], and Pen2 and Aph-1aL were cloned into the *Kpn*I and *Xba*I sites of pBEVY-L [50]. DNA coding for C55-Gal4p (APP 672–726 fused to the SUC2 signal peptide sequence and Gal4p) was cloned into the *Bam*HI and *Eco*RI sites of p426ADH [51]. These recombinant plasmids were transferred to *Saccharomyces cerevisiae* strain PJ69-4A (*MATa trp1-901 leu2-3, 112 ura3-52 his3-200 gal4Δ gal80Δ LYS2::GAL1-HIS3 GAL2-ADE2 met2::GAL7-lacZ*) [52]. Substrate cleavage was assessed by monitoring the expression of reporter genes. The expression of *HIS3* and *ADE2* was estimated based on colony growth on minimal SD agar medium lacking leucine, tryptophan, histidine, adenine, and uracil (SD-LWHUA). β-Galactosidase was assayed using *O*-nitrophenyl β-d-galactopyranoside [26]. Exponentially proliferating cells (1 × 10^7^ cells) were lysed using glass beads in 30 µL of lysis buffer (20 mM Tris-HCl, pH 8.0, 10 mM MgCl_2_, 50 mM KCl, 1 mM EDTA, 5% glycerol, and 1 mM dithiothreitol [DTT]) containing a protease inhibitor mixture. The cell lysate was centrifuged at 15,000× rpm for 10 min, and the β-galactosidase activity and protein concentration (Bradford protein assay; Bio-Rad, Hercules, CA, USA) were measured in the supernatant.

### 4.2. Random Mutagenesis by Inverse PCR

Random mutations were introduced at APP V721/M722 or L723 of the APP T714I FAD mutant by inverse PCR. Briefly, V721/M722 or L723 were mutated using the following primers with random nucleotides: V721/M722-S, 5′-CTGAAGAAGAAAACTAGTATGAAGCT-3′ and V721/M722-AS, 5′-NNNNNNCAAGGTGATGACGATCACTATCG-3′, or L723-S, 5′-AAGAAGAAAACTAGTATGAAGCTACTG-3′ and L723-AS, 5′-NNNCATCACCAAGGTGATGACGATCA-3′ (N stands for mixed bases). PCR was performed in 20 μL of the solution (50 ng of template DNA, 2.5 mM dNTPs, and 10 µM of each primer in 1× *PfuUltra* buffer) with 1 unit of *PfuUltra* High-Fidelity DNA Polymerase and the following cycles: 95 °C for 30 s, 95 °C for 30 s, 55 °C for 1 min, and 68 °C for 24 cycles. PCR products were phosphorylated with T4 polynucleotide kinase, ligated with T4 DNA ligase, and transformed into *S. cerevisiae* PJ69-4A. Transformants were screened on selection medium plates SD-LWHUA (Table 1). Plasmid DNA was isolated from yeast colonies and mutations were identified using DNA sequencing. Other site-specific mutations were introduced using a QuikChange mutagenesis kit (Stratagene, La Jolla, CA, USA).

### 4.3. Aβ Production in Chinese Hamster Ovary Cells

Aβ production was analyzed in CHO cells introduced with WT and a C99 mutant that contained amino acids 672–770 of human APP770, which was inserted into the *Nhe*I/*Bst*XI site of pcDNA(-) 3.1 neo plasmid [53]. Cells were cultured in Dulbecco’s modified Eagle’s medium (DMEM) supplemented with 10% fetal bovine serum and 1% antibiotics (penicillin and streptomycin), and maintained at 37 °C and 5% CO_2_ in a tissue culture incubator. Cells were transfected with WT or mutant C99-pcDNA plasmid using Lipofectamine 2000 (Invitrogen, Waltham, MA, USA) according to the manufacturer’s instructions. After 24 h, the media were replaced by DMEM that lacked serum and antibiotics. After 48 h of incubation, the conditioned media and cells were recovered.

Secreted Aβ40 and Aβ42 were quantified in the media using two-site ELISAs (Human Amyloidβ [1–40] Assay Kit [#27713; IBL, Gunma, Japan] and Human Amyloidβ [1–42] Assay Kit [#27711; IBL]). The amounts of Aβ secreted were also measured in the media and cells, which were subjected to immunoblot analysis after gel electrophoresis [32]. Aβ species were specifically detected on Tris/Tricine/8 M urea gels at pH 8.45 or 8.65. The γ-secretase-specific inhibitor N-[N-(3,5-difluorophenacetyl)-l-alanyl]-S-phenylglycine t-butyl ester (DAPT) was included in the conditioned media as a control. The blots were developed using an ECL system and the intensities of the bands were quantified using an LAS-4000 image analyzer (Fujifilm, Japan).

### 4.4. Antibodies

The antibodies used for immunoblotting were polyclonal antibodies against the Gal4 activation domain (Anti-GAL4; Sigma-Aldrich, St. Louis, MO, USA), polyclonal antibodies against human PS1 loop (GIL3) [25,27,33,34], monoclonal antibodies against Aβ N-terminus (82E1; IBL), and monoclonal antibodies against tubulin (DM1A; Novus Biologicals, Englewood, CO, USA).

## Figures and Tables

**Figure 1 ijms-24-03970-f001:**
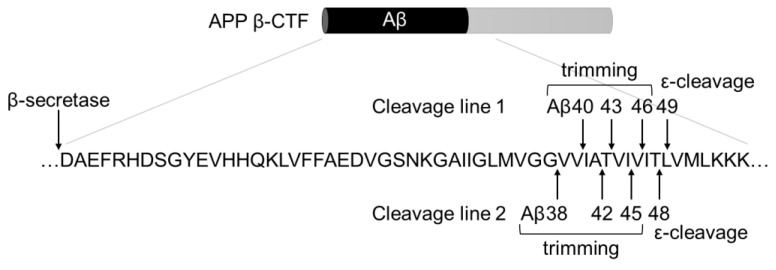
Cleavage model of APP β-CTF by γ-secretase. APP is cleaved by β-secretase to form β-CTF, and β-CTF is cleaved by γ-secretase. γ-Secretase is cleaved sequentially through ε-cleavage, followed by trimming. Two types of cleavage lines exist: one produces Aβ49 through ε-cleavage, followed by Aβ46 → 43 → 40 trimming (cleavage line 1), and the other produces Aβ48 through ε-cleavage, followed by Aβ45 → 42 → 38 trimming (cleavage line 2).

**Figure 2 ijms-24-03970-f002:**
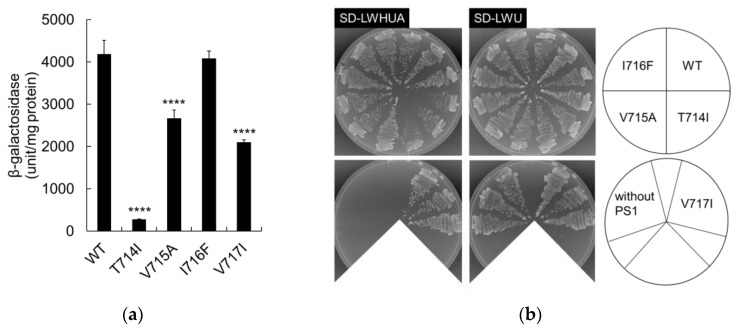
β-Galactosidase activity and growth of APP FAD mutants. Wild-type (WT) or FAD-mutant APPC55-Gal4p was introduced into yeast with PS1, NCT, Pen2, and Aph-1aL. Gal4p cleaved from APPC55-Gal4p activates the *lacZ* gene, which encodes β-galactosidase. (**a**) β-Galactosidase activity of APP FAD mutants. One unit of β-galactosidase activity corresponds to 1 nmol of *O*-nitrophenyl β-galactoside hydrolyzed per minute (expressed as units/mg lysate protein). Activity was normalized to control cells without PS1 (25 units/mg protein). Representative results from three independent assays are shown with standard deviations. Statistical analyses were performed by one-way analysis of variance (ANOVA) and Dunnett’s multiple comparison test. Asterisks indicate *p* < 0.0001 (****) compared to the WT; (**b**) The growth of cells expressing WT or FAD-mutant APP was examined after incubation at 30 °C for 3 days in selection media (SD-LWHUA). Three independent clones were evaluated for each strain. The growth of the mutants in medium containing histidine and adenine (SD-LWU) was comparable to that of the WT.

**Figure 3 ijms-24-03970-f003:**
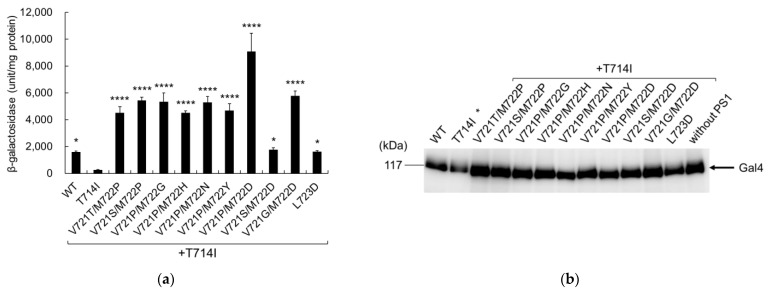
β-Galactosidase activity of secondary mutations in APP T714I. Wild-type (WT) or mutant APPC55-Gal4p was introduced into yeast with PS1, NCT, FLAG-Pen2, and Aph-1aL-HA. (**a**) β-Galactosidase activity was assayed for each cell expressing WT or mutant APPC55-Gal4p. β-Galactosidase activity was normalized to that of the control cells without PS1 (56 units/mg protein). Representative results from three independent assays are shown with standard deviations. Asterisks indicate *p* < 0.0001 (****) or *p* < 0.05 (*) compared to T714I; (**b**) Expression of WT or mutant APP in yeast. Cell lysates from the WT or mutant APP were analyzed using immunoblot analysis and an antibody against the activation domain of Gal4. The arrow indicates the position of Gal4. An asterisk indicates that this particular lane (T714I) is not characteristic. The expression level of APP T714I is similar to WT (Appendix A).

**Figure 4 ijms-24-03970-f004:**
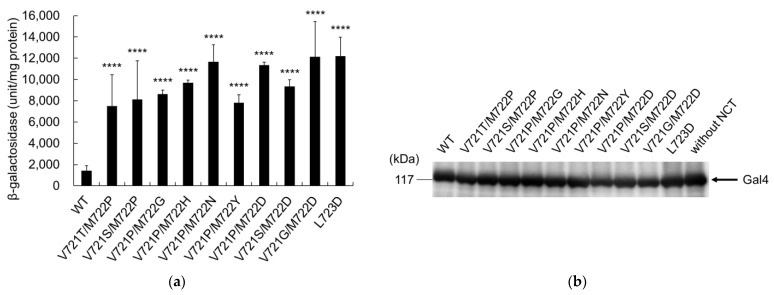
β-Galactosidase activity of APP with only secondary mutations. (**a**) β-Galactosidase activity of the cells expressing either WT or mutant APPC55-Gal4p was assayed as in Figure 3. Activity was normalized to that of control cells without nicastrin (30 units/mg protein). Asterisks indicate *p* < 0.0001 (****) compared to the WT; (**b**) Expression of WT or mutant APP in yeast. Cell lysates from the WT or mutant APP were analyzed using immunoblot analysis, as in Figure 3. The arrow indicates the position of Gal4.

**Figure 5 ijms-24-03970-f005:**
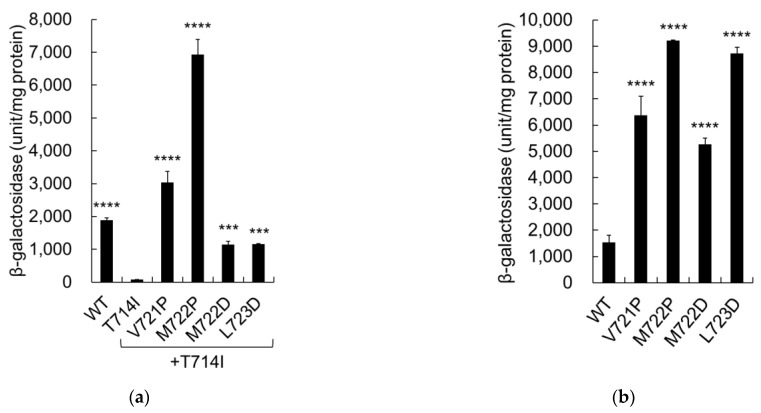
β-Galactosidase activity of APP proline and aspartate mutants. (**a**–**d**) β-Galactosidase activity was assayed for each cell expressing either WT or mutant APPC55-Gal4p, as in Figure 3. Activity was normalized to that of the control cells without PS1 (61 units/mg protein for (**a**,**c**) and 63 units/mg protein for (**b**,**d**)). Asterisks indicate *p* < 0.0001 (****), *p* < 0.001 (***), *p* < 0.01 (**), or *p* < 0.05 (*) compared to the control (T714I for (**a**,**c**) or the WT for (**b**,**d**)); (**e**,**f**) Expression of WT and mutant APP in yeast. Cell lysates were analyzed using immunoblot assays, as in Figure 3. The arrow indicates the position of Gal4.

**Figure 6 ijms-24-03970-f006:**
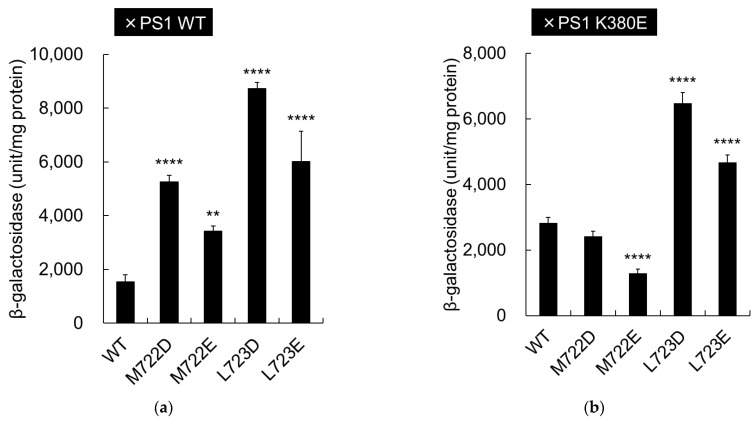
Structural basis for the activation of APP aspartate and glutamate mutations. (**a**) β-Galactosidase activity was assayed for each cell expressing WT or mutant APPC55-Gal4p with PS1 WT. Activity was normalized to that of the control cells without PS1 (63 units/mg protein); (**b**) β-Galactosidase activity was assayed for each cell expressing WT or mutant APPC55-Gal4p with PS1 K380E. Activity was normalized to that of the control cells without PS1 (88 units/mg protein). Asterisks indicate *p* < 0.0001 (****) and *p* < 0.01 (**) and *p* < 0.05 (*) compared to the WT; (**c**) The expression of WT or mutant APP or PS1 in yeast. Cell lysates were analyzed by immunoblotting using an antibody against the activation domain of Gal4 or the PS1-loop region (GIL3). An asterisk denotes nonspecific bands. Arrows indicate the positions of Gal4, PS1, and the PS1 C-terminal fragment (CTF); (**d**) The position of APP and PS1 mutations. The structure of human PS1-APP (Protein Data Bank code: 6IYC) is shown in yellow (APP) and blue (PS1). PS1 K380 is highlighted in green.

**Figure 7 ijms-24-03970-f007:**
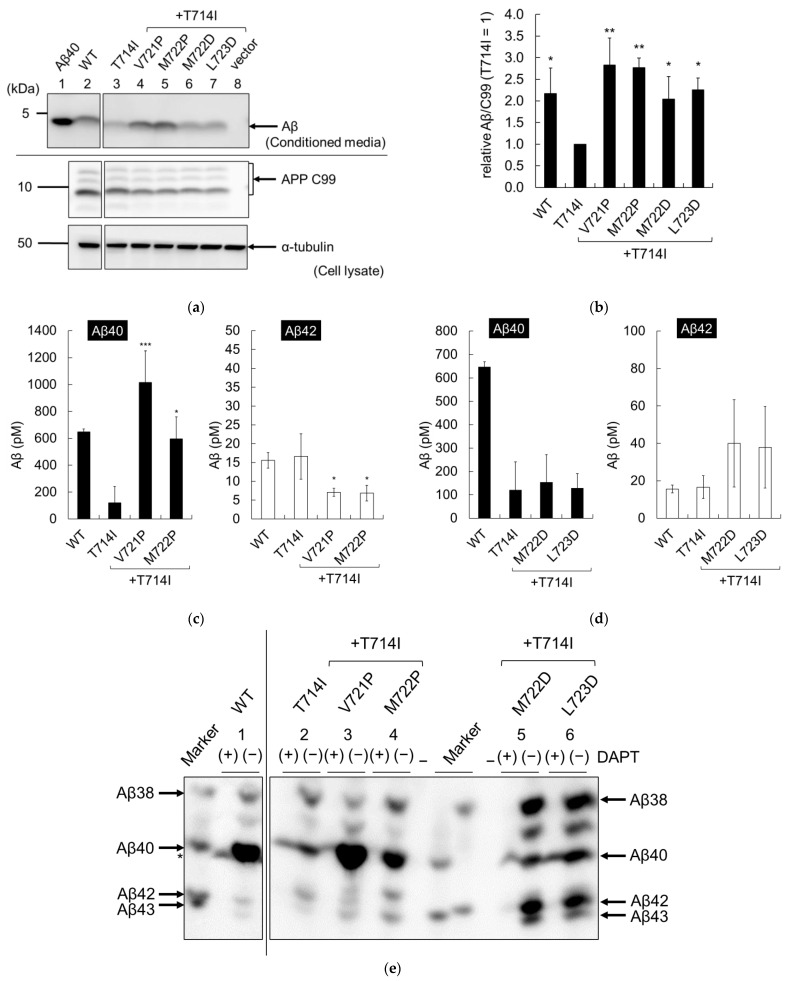
Aβ production in Chinese hamster ovary (CHO) cells containing APP secondary mutations and T714I. CHO cells were transfected by lipofection with WT or mutant APPC99, and an empty vector. (**a**) Secreted Aβ in the media and APP C99 in cell lysates were detected using 82E1. We also detected α-tubulin using a DM1A antibody; (**b**) The cleavage rate (Aβ/C99) was quantified form immunoblots; (**c**,**d**) Aβ secreted from the recovered media was quantified using two-site ELISA with antibodies against Aβ40 (left panel) and Aβ42 (right panel). For quantification, each value was subtracted from the vector control (349 pM for Aβ40 and 10 pM for Aβ42); (**e**,**f**) The recovered media were also assessed using urea/SDS-PAGE analysis at pH 8.45 (**e**) or pH8.65 (**f**) and secreted Aβ was detected using 82E1. Aβ38, Aβ40, Aβ42, and Aβ43 peptides (Peptide Institute, Japan) were loaded as controls. Each mutant was controlled by adding 10 µM DAPT, which is a γ-secretase inhibitor. An asterisk denotes nonspecific bands. Numbers 1–6 denotes lane numbers; (**g**,**h**) The proportions of the Aβ fragments Aβ38 (diagonal striped), Aβ40 (black), Aβ42 (white), and Aβ43 (horizontal striped) were quantified from immunoblots. All quantifications were conducted using three independent assays and are shown with standard deviations. Asterisks indicate *p* < 0.0001 (****), *p* < 0.001 (***), *p* < 0.01 (**), or *p* < 0.05 (*) compared to T714I.

**Figure 8 ijms-24-03970-f008:**
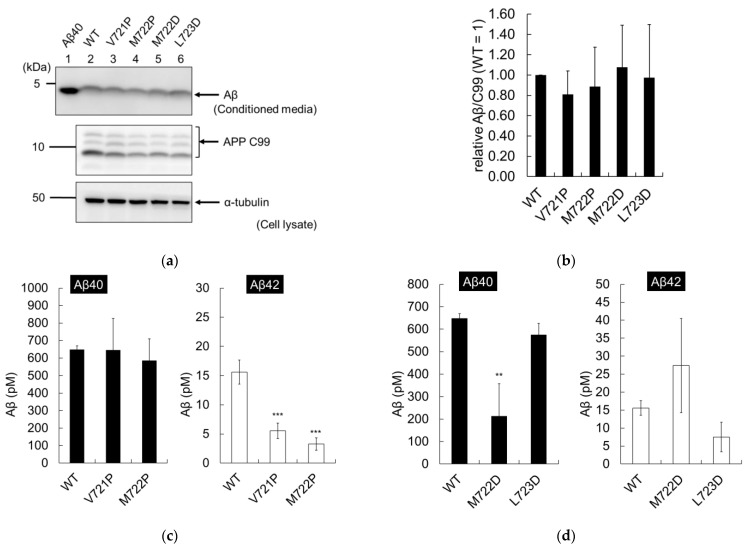
Aβ production by secondary APP mutations in Chinese hamster ovary (CHO) cells. CHO cells were transfected with WT or mutant APPC99, and an empty vector through lipofection. (**a**) Aβ secreted in the media, and APP C99 or α-tubulin in cell lysates, were analyzed by immunoblotting, as shown in Figure 7; (**b**) Cleavage rates (Aβ/C99) were quantified using three independent assays and are presented with standard deviations; (**c**,**d**) Aβ secreted from the recovered media was quantified by two-site ELISA using three independent assays and is presented with standard deviations, as shown in Figure 7; (**e**,**f**) The recovered media were also analyzed by urea/SDS-PAGE at pH 8.45 (**e**) or pH8.65 (**f**) and secreted Aβ was detected using 82E1. Numbers 1–6 denotes lane numbers; (**g**,**h**) The proportions of each Aβ fragment were determined using four independent assays and are shown with the standard deviations. Asterisks indicate *p* < 0.001 (***) or *p* < 0.01 (**) and *p* < 0.05 (*) compared to the WT.

**Figure 9 ijms-24-03970-f009:**
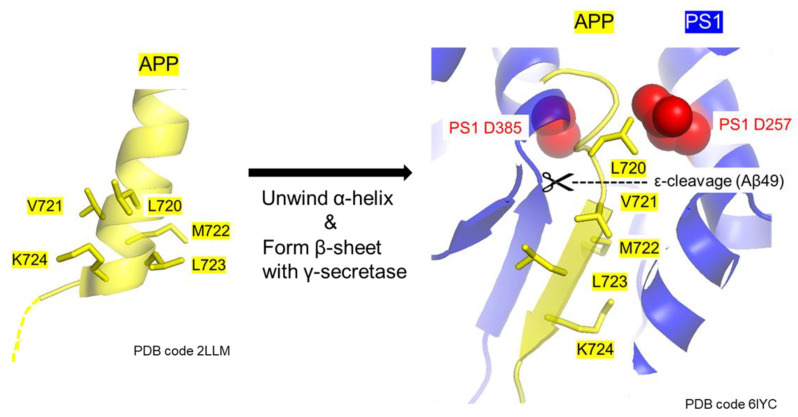
Structure-based models of secondary APP mutations. Structure-based models of the transmembrane domain of human APP (Protein Data Bank code 2LLM) and the transmembrane core of human APP recognized by γ-secretase (Protein Data Bank code 6IYC) are shown. These models, i.e., PS1 (blue) and APP (yellow), are indicated by a sphere containing the residues Asp257 and Asp385 (red). APP _721_VMLK_724_ forms an α-helix that is unwound and forms a β-sheet with γ-secretase when APP is cleaved. The secondary proline mutations (V721 and M722) may facilitate this unwinding.

**Table 1 ijms-24-03970-t001:** Growth of secondary mutants that restored cleavage of the APP T714I FAD mutant. The growth of cells containing APP mutations was analyzed on SD-LWHUA media after 3 days at 30 °C. All cells contained NCT, Aph-1aL-HA, and FLAG-Pen2 with or without PS1. Cells (3.0 × 10^3^) were screened for V721/M722 mutants and 6.0 × 10^2^ cells were screened for L723 mutants. Each mutant was isolated from one clone. “++” represents wild-type growth with cells forming > 1 mm colonies; “−” represents no growth.

APP Mutants	Growth with PS1 WT	Growth without PS1
WT	++	
APP T714I	−	
APP T714I, PCR mutation		
V721T/M722P	++	− *
V721P/M722G	++	−
V721S/M722P	++	−
V721P/M722D	++	− *
V721S/M722D	++	−
V721P/M722H	++	−
V721G/M722D	++	−
V721P/M722Y	++	−
V721P/M722N	++	−
L723D	++	−

* very small colonies.

**Table 2 ijms-24-03970-t002:** Summary of the effects of APP aspartate and glutamate mutants. In Figure 6a,b, each APP WT was defined as one. This table indicates how many times more active the aspartate mutants (M722D and L723D) and glutamate mutants (M722E and L723E) were than the WT. The value is expressed as the average of three independent assays ± 95% confidence interval.

	APP	WT	M722D	M722E	L723D	L723E
PS1						
WT		1.00 ± 0.19	3.42 ± 0.16	2.23 ± 0.13	5.67 ± 0.18	3.91 ± 0.83
K380E		1.00 ± 0.07	0.86 ± 0.06	0.45 ± 0.06	2.29 ± 0.13	1.65 ± 0.09

**Table 3 ijms-24-03970-t003:** Summary of the proportions of Aβ species produced by APP WT, T714I, and secondary mutants with T714I in CHO cells. Percentages are expressed as the average of three independent assays ± 95% confidence interval.

	WT	T714I	*T714I*/ V721P	*T714I*/ M722P	*T714I*/ M722D	*T714I*/ L723D
Short Aβs (Aβ38 + Aβ40)	91.4 ± 4.7	78.5 ± 5.1	94.3 ± 1.3	85.8 ± 3.1	59.5 ± 7.3	67.7 ± 13.1
Long Aβs (Aβ42 + Aβ43)	8.6 ± 4.7	21.5 ± 5.1	5.7 ± 1.3	14.2 ± 3.1	40.5 ± 7.3	32.3 ± 13.1
Cleavage line 1; Aβ49 → 46 → 43 → 40 (Aβ40 + Aβ43)	71.8 ± 8.6	34.4 ± 5.3	91.3 ± 4.5	61.4 ± 2.1	24.2 ± 3.4	26.3 ± 0.9
Cleavage line 2; Aβ48 → 45 → 42 → 38 (Aβ38 + Aβ42)	28.2 ± 8.6	65.6 ± 5.3	8.7 ± 4.5	38.6 ± 2.1	75.8 ± 3.4	73.7 ± 0.9

**Table 4 ijms-24-03970-t004:** Summary of the proportions of Aβ species produced by APP WT and secondary mutants in CHO cells. Percentages were expressed as the average of four independent assays ± 95% confidence interval.

	WT	V721P	M722P	M722D	L723D
Short Aβs (Aβ38 + Aβ40)	88.8 ± 6.1	92.5 ± 3.8	93.1 ± 3.2	71.0 ± 14.6	92.6 ± 7.7
Long Aβs (Aβ42 + Aβ43)	11.2 ± 6.1	7.5 ± 3.8	6.9 ± 3.2	29.0 ± 14.6	7.4 ± 7.7
Cleavage line 1; Aβ49 → 46 → 43 → 40	70.9 ± 6.3	90.4 ± 2.0	88.6 ± 3.3	46.3 ± 7.9	83.1 ± 5.9
Aβ40 + Aβ43					
Cleavage line 2; Aβ48 → 45 → 42 → 38	29.1 ± 6.3	9.6 ± 2.0	11.4 ± 3.3	53.7 ± 7.9	16.9 ± 5.9
Aβ38 + Aβ42					

## Data Availability

Not applicable.

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
