# Peer review of "Specific Mutations near the Amyloid Precursor Protein Cleavage Site Increase γ-Secretase Sensitivity and Modulate Amyloid-β Production"

_ijms, 2023, doi:10.3390/ijms24043970_

Round 1
Reviewer 1 Report
The study by Suzuki et al. is interesting but it lacks the clarity and scientific rigor for publication in the current format. The figure captions as well as figure quality (especially blots in Figure 7 and 8)need to be improved. While authors performed beta-galactosidase activity analysis, they do not perform or even discuss similar experiments for gamma-secretase activity in CHO cells. I believe those experiments would be extremely important to justify the claims that authors are making about the mutations. In addition, the manuscript would improve substantially if authors performed other supporting experiments. For ex, FRET-based fluorescence lifetime imaging to monitor APP-PS1 interactions. Overall, I believe the manuscript is not currently mature enough to be accepted for publication.

Reviewer 2 Report
The article “Specific Mutations near the Amyloid Precursor Protein Cleavage Site Increase γ-Secretase Sensitivity and Modulate Amyloid-β Production” by Suzuki et al. describes the selection of secondary mutations in residues located next to the gamma-secretase endopeptidase-like cleavage site that rescue the effect of the FAD mutation T714I. The reconstituted yeast selection system has been described in an earlier study from the same corresponding author. Interestingly, all the secondary mutations had a change to either P or D. In the case of P substitutions, the cleavage-facilitating effect responsible for the T714I rescue is likely due to a disruption of an alpha-helix. The shift towards one of the two conventional gamma-secretase trimming pathways and, possibly, a novel trimming pathway may be explained by a disruption of the intermolecular beta-sheet between APP and PS1. The effect of D substitutions (and also generated negatively charged E substitutions) is possibly due to the interaction with the PS1 K380 residue. In general, the study is well thought through and pretty extensive, and I strongly recommend publishing this paper.
However, several concerns / questions have to be addressed before the manuscript can be accepted. In addition, some minor text corrections are needed.
1. Fig. 2B. No effect of mutations in positions 715 and 717 can be seen on the image provided even though a statistically significant effect is seen in Fig. 2A. Is it possible that a weak effect is lost because the incubation time is too long (3 days)? I suggest providing an image after 2 days of incubation, which may still show a growth difference.
2. Table 1 indicates a “-“, which is described as “no growth” for the T714I mutation in the presence of WT PS1. This clearly contradicts data shown in Fig. 2B where such cultures have a moderately reduced growth rate. Please, explain or revise accordingly.
3. Fig. 2A vs Fig. 3A and all subsequent figure panels showing beta-galactosidase activity. In Fig. 2A the activity in transformants carrying a WT APP fragment construct is ~4,000, whereas it is lower than 2,000 in other similar graphs. The significance for WT vs T714I also changes from p<0.001 to p<0.05. While beta-galactosidase activity can vary from experiment to experiment, this is a huge difference. Can a strangely high WT result shown in Fig. 2B explain the difference between WT and 715 and 717 mutants, which is not seen in Fig. 2B? Please, provide an explanation or redo.
4. Fig. 3B. The expression level for the T714I construct appears to be considerably lower compared to WT and especially some other constructs (looks like up to 5-fold difference). Luckily, this is not the case in Fig. 5C. I assume that the conclusion about “similar levels of expression” was made based on multiple gels. Please, put an asterisk and make a special note that this particular lane is not characteristic, and provide quantification for several gels for WT and T714I.
5. Fig. 6B. I did not understand why the activity is normalized to different controls within the same graph for WT and K380 PS1 while the legend says that both controls lacked PS1 altogether. Is it correct that data comes from experiments not done in parallel? If this is the case, this is a major problem, especially considering comment 3 above. Please, explain and redo, if experiments were not parallel.
6. Fig. 6B. The explanation for probability asterisks in the legend does not correspond to the figure. As far as I understand, there should be only one probability for each set of WT and K380, either above WT or above K380. From what I understand the asterisks mentioned in the legend are above the WT bars. What do asterisks above the K380 bars mean? In the case they show difference between APP WT and mutants (Table 2 data), this is very confusing and not explained in the legend. Please, revise the figure. Probably just by removing the second set of asterisks and adding confidence intervals to Table 2.
7. Table 2. Please, add confidence intervals.
8. Lines 211-212. “The degree of activation observed with APP aspartate and glutamate mutations was low with PS1 K380E compared to PS1 WT (Table 2)”. Is it correct to say “low, none, or even negative”?
9. Lines 273-276. The analysis of the A-beta42 / A-beta40 ratio is a bit confusing because it involves both the efficiency of trimming and the choice of the trimming pathway. Would it be more appropriate to analyze A-beta42 / A-beta38 and A-beta43 / A-beta40, or A-beta42 + A-beta43 / A-beta38+A-beta40? Indeed, the authors do a great job distinguishing between trimming activity and pathway choice later on, but it was pretty hard to move along this section.
10. For all ratios and percentages based on Fig. 7 and 8 data. Please, clarify which differences are statistically significant. Maybe a table with confidence intervals could help.
Minor issues:
1. Lines 14-15 “..mutations that activate and restore the loss of function”. Consider revising language.
2. Line 17 “… restore the FAD mutation…” . Consider revising language.
3. Line 38. While it is well established that accumulation of longer A-beta fragments is the cause of AD, some investigators in the field would argue that AD is caused by events preceding or occurring in parallel with the accumulation of fibrils and plaques. Consider revising the statement.
4. Fig 2B and further. Why is such nomenclature for the SD-LWHUAde is chosen? Either use a single letter nomenclature for all media components, or use a three-letter nomenclature for both adenine and uracil (and a single letter nomenclature for amino acids). Analogously, “His and adenine” seems to be a strange combination in the figure legend (line 109).
5. For all figure legends. Panel names ((a), (b), etc.) are placed differently in every legend – right after the title, in the middle, in the end, or even twice. This is confusing. Please, fix according to journal rules.
6. Lines 298-300 seem to belong to the previous paragraph.
7. Line 392. “Glutamate mutations may not have been identified during screening because their recovery levels may have been lower than those of the aspartate mutations”. Has this experiment been done? If yes, it is definitely worth including.
Reviewer 3 Report
Very interesting manuscript with good quality data.
Round 2
Reviewer 1 Report
The authors have addressed all the comments and therefore the current form of manuscript can be accepted for publication.
Reviewer 2 Report
I am really impressed with the revisions made to address my comments, as well as the other Reviewer's critiques. In my opinion, the current version of the manuscript is much clearer and more streamlined. I recommend the acceptance in the present form.